# Appropriateness of high-priority criteria and safety of endoscopy procedures during the COVID-19 lockdown

**Dalia Morales-Arraez**[1], **Anjara Hernández**[1], **Alberto Hernández-Bustabad**[1], **Carla Amaral**[1], **Cristina Reygosa**[1], **David Nicolás-Pérez**[1], **Antonio Zebenzui Gimeno-García**[1], **Manuel Hernández-Guerra**[1,2]*

**1** Gastroenterology and Hepatology Department, Hospital Universitario de Canarias, Tenerife, Spain,
**2** Instituto Universitario de Tecnologías Biomédicas CIBICAN, Departamento de Medicina Interna, Psiquiatría y Dermatología, Universidad de La Laguna, Tenerife, Spain

\* mhernand@ull.edu.es

## Abstract

### Background

During the coronavirus-19 disease (COVID-19) pandemic, gastroenterology guidelines recommended the suspension or reduction of non-urgent endoscopy. We aimed to assess the appropriateness and safety of endoscopic activity during the pandemic first wave lockdown using European Society of Gastrointestinal Endoscopy (ESGE) recommendations.

### Methods

We identified scheduled patients from the onset of the lockdown in Spain since March 16, 2020) to April 14, 2020. Daily hospital COVID-19-related burden was also registered. A similar cohort from a period immediately before the lockdown was studied (pre-lockdown cohort) to compare appropriateness.

### Results

454 endoscopy procedures were performed during the studied period, comprising a 49.7% reduction compared to the pre-lockdown cohort (n = 913). There was a significant increase in ESGE high-priority indications (62.1% vs. 45.6%, p<0.001) associated with an increase in relevant endoscopic findings (p = 0.006), advanced neoplasia/cancer (p = 0.004) and cancer detection rate (p = 0.010). There were no differences in the rate of admissions or infection among scheduled patients in the lockdown cohort. None of the staff members tested positive for COVID-19 in the 7 days after the adoption of protective measures.

### Conclusion

A prioritized endoscopic activity is not associated with higher contagion after adopting protective measures. In addition, a triage of procedures that follow the ESGE criteria increases the rate of relevant endoscopic findings. These considerations may reduce the impact of the delays of diagnosis after the pandemic.

**Data Availability Statement:** Individual data cannot be shared due to confidentiality restrictions

imposed by the Clinical Research Ethics Committee of the Hospital Universitario de Canarias (CRECHUC). Data are available upon request from CRECHUC (ceticohuc.scs@gobiernodecanarias. org), for researchers who meet the criteria for access to confidential data.

**Funding:** This study was supported in part by grants from Fondo Europeo de Desarrollo Regional (FEDER). Dr. M. Hernandez-Guerra is the recipient of a grant from Instituto de Salud Carlos III (PI19/ 01756). The funders had no role in study design, data collection and analysis, decision to publish, or preparation of the manuscript.

**Competing interests:** The authors have declared that no competing interests exist.

**Abbreviations:** COVID-19, Coronavirus-19 disease; ERCP, Endoscopic retrograde cholangiopancreatography; ESGE, European Society of Gastrointestinal Endoscopy; EUS, Endoscopic ultrasound; FOBT, Fecal occult blood test; GI, Gastrointestinal; PEG, Percutaneous endoscopic gastrostomy.

## Introduction

After the worldwide spread of the highly contagious coronavirus-19 disease (COVID-19) [1, 2], the World Health Organization declared a global pandemic on March 11[th], 2020 [3].

Coronavirus causes a flu-like syndrome and spreads through respiratory droplets making upper endoscopic procedures a high-risk maneuver [4]. The virus has also been isolated in the stools, and this may represent another possible mode of transmission [5–9]. Therefore, gastro-intestinal (GI) endoscopy units are considered high-risk COVID-19 contagion areas [10]. Accordingly, scientific societies have recommended drastically reducing routine activity in order to allocate resources, redefine priorities to balance risk-benefit, and to implement strict preventive measures for protecting hospital professionals and patients [11–14].

However, most of these recommendations were based on low or very low certainty of evidence. Therefore, without taking into account regional risk areas or number of cases in local hospitals and proper assessment of the risk of contagion, these recommendations may not constitute appropriate measures in all cases.

In fact, a paper from Italy showed that the rate of contagion among patients after attending an endoscopic procedure during the outbreak was low [15], and despite reported high infections rates among health care workers [16], a low rate on infection was observed among endoscopy unit professionals.

In contrast, cancellation or reduced activity in endoscopy units, specifically relating to non-high-risk procedures, such as colonoscopies, may have dramatic consequences including oncologic delays in diagnosis, especially in an open-access public health setting with waiting lists that exceed more than six months [17–19]. Additionally, delays may lead more patients to the Emergency Department during the outbreak and increase admissions in hospital for progressive symptoms.

There is a need to provide evidence of the benefit of maintaining endoscopic activity according to predefined priority criteria, as well as the risk of contagion in relation to a specific burden scenario. This may be pivotal for future evidence-based recommendations and clinical guidance, which will be particularly useful in managing future waves and lockdowns related to COVID-19 pandemic.

Therefore, our observational study aimed to assess the appropriateness of maintaining endoscopy activity according to triage priority criteria and safety during the COVID-19 pandemic outbreak in our public health tertiary hospital setting during the lockdown.

## Materials and methods

This study was performed in the open-access endoscopy unit at the Hospital Universitario de Canarias. This is a tertiary referral endoscopy unit for the northern area of Tenerife island (Canary Islands, Spain), which has 485,000 inhabitants.

The study was conducted following the ethical principles of the Declaration of Helsinki (October 2013), and approval was obtained from the Clinical Research Ethics Committee of the Hospital Universitario de Canarias (code: CHUC_2020_28 [END_COVID]), e-mail: ceti-cohuc.scs@gobiernodecanarias.org).

The informed consent was not required due to the retrospective design of the study and we obtained consent waiver from the Ethics Committee.

### Measures during the COVID-19 pandemic

Since March 16, 2020, and during the lockdown announce in Spain, a restructure of activity was performed in our Gastrointestinal and Hepatology Department including the daily number of hospital and outpatient gastrointestinal endoscopy procedures in planned agenda. The

procedures were scheduled according to the priority of clinical indication after a case-by-case assessment by physicians from the Endoscopy Unit of our hospital, checking clinical indication and medical record, and according to National Institute for Health and Care Excellence (NICE) criteria [17] for lower gastrointestinal (GI) endoscopy and Asociación Española de Gastroenterologia (AEG) guidelines [20] for upper GI endoscopy. In addition, several protective measures were adopted (S1 Table) according to scientific society guidelines and position statements [11–14].

### Primary and secondary outcomes of the study

The primary outcome was the evaluation of the appropriateness of endoscopy procedures during the COVID-19 pandemic lockdown as the rate of high-priority procedures according to the European Society of Gastrointestinal Endoscopy (ESGE) criteria. As secondary outcomes we evaluated the effectiveness (rate of relevant endoscopic findings, advanced neoplasia/cancer and cancer detection), non-scheduled visits to the hospital and patients and endoscopy staff safety.

### Endoscopic activity, appropriateness, effectiveness, visits to the hospital and safety evaluation

For the purpose of the study we compared a cohort from the start of the lockdown on March 16, 2020 to April 14, 2020 (lockdown cohort) to a cohort from January 13, 2020 to February 9, 2020 (pre-lockdown cohort), which was prior to the first COVID-19 case in Tenerife island on February 23, 2020 and within a normally scheduled activity of our Endoscopy Unit.

All registered endoscopy procedures (inpatient and outpatient) were recorded and categorized in upper GI endoscopy including gastroscopy, endoscopic retrograde cholangiopancreatography (ERCP), upper endoscopic ultrasound (EUS) and percutaneous endoscopic gastrostomy (PEG), and in lower GI endoscopy including colonoscopy, rectosigmoidoscopy and lower EUS. Videocapsule and enteroscopy were not classified in any of these groups.

To evaluate the appropriateness, we focused on elective scheduled procedures and excluded those to be performed immediately (mandatory) that were related to patients admitted into hospital or that were urgent procedures during on-call duty. Clinical indications were identified from medical records and were classified into high and low-priority endoscopy procedures according to the ESGE criteria [12]. Severe anemia was defined as hemoglobin ≤ 11 gr/dl in men and ≤ 10 gr/dl in women [17].

Non classifiable (not referenced in the ESGE guideline) clinical indications including constitutional syndrome, general colorectal cancer screening without fecal occult blood test (FOBT), screening and surveillance of gastroesophageal varices and gastric ulcer, celiac disease suspicion, and emetic syndrome were grouped as "not classifiable".

The effectiveness of appropriately selected criteria was assessed according to predefined endoscopic findings that were considered relevant due to the prognosis, symptoms, and whenever further effective medical and endoscopic therapies could be assured (S2 Table). In addition, we separately evaluated advanced neoplasia (defined as a polyp larger than 10 mm and/or with high-grade dysplasia) and cancer, and cancer at the endoscopic procedure. As an indirect marker of appropriateness, we also registered visits to the Emergency Department and hospital admissions related to the medical indication for endoscopy until 31st December 2020.

For safety assessment, using medical records, we retrospectively identified the endoscopist (n = 21), anesthetist (n = 2), and nurse and auxiliary nurse (n = 20) attending the procedures, as well as RT-PCR and serology positive results for SARS-CoV-2 using immunochromatography of staff and schedule patients in the lockdown cohort until April 25, 2020 and from the

date of the scheduled endoscopy procedure to 20 days thereafter, respectively. Patients who underwent endoscopy procedure were not specifically tracked for COVID-19 infection, and only laboratory records were retrospectively checked to find positive results. To investigate the risk of COVID-19 infection, attendees to the procedure were compared to non-attendees (cases of no show and cancellation).

In the lockdown period, RT-PCR was performed only in symptomatic subjects and not as a routine screening before the endoscopy procedure. In the same way, for endoscopy staff, the COVID-19 tests were performed in case of a positive result was detected in the unit and in the context of an opportunist screening performed in our institution on 25th April 2020.

## Burden related to COVD-19

To provide references of the burden of the pandemic in our hospital during the study period, the number of hospital admissions in the Gastroenterology and Hepatology Department, COVID-19 admissions, and the number of visits to the Emergency Department were registered daily. In addition, the total hospital allocation (n = 688) was calculated from the total hospital bed capacity.

## Statistical analysis

The chi-squared test was used to assess qualitative variables, and Student's t-test or the Mann-Whitney U test was used for continuous variables, as appropriate. Spearman's Rho was used to explore correlations. The relative risk (RR) with confidence interval (CI) were calculated for findings. P values $<0.05$ were considered statistically significant. SPSS v 26.0 was used for all statistical analyses.

## Results

### Endoscopic activity

A total of 1367 endoscopy procedures were performed in our catchment area, 913 in the pre-lockdown cohort (49.2% male, 63.2 ± 15.4 years) and 454 in the lockdown cohort (57.3% male, 64.2 ± 14.3 years) (Fig 1). The number of procedures in the lockdown cohort was 51.1% of the total planned scheduled agenda for the period (n = 889); this was a 49.7% reduction in procedures compared to the pre-lockdown cohort.

The number of outpatient procedures in the pre-lockdown cohort and lockdown cohort was 717 and 319, respectively. The percentage of procedures scheduled as outpatient explorations in the pre-lockdown cohort was slightly higher than that in the lockdown cohort (78.5% vs. 70.3, p = 0.001). Regarding urgent endoscopy, the number of procedures was 196 in the pre-lockdown cohort and 135 in the lockdown cohort (Fig 2).

### Appropriateness of outpatient endoscopy procedures during the lockdown

Table 1 shows the medical indications for outpatient endoscopy procedures during the pre-lockdown and lockdown cohort categorized according to ESGE classification in high and low-priority indications. S3 Table shows the clinical indications for inpatients.

Regarding outpatient endoscopy procedures, there was a significant increase in the percentage of high-priority indications in the lockdown cohort compared to the pre-lockdown cohort (62.1% vs. 45.6%, p<0.001). This increase was irrespective of upper (54.3% vs. 45.2%, p = 0.145) and lower (65.2% vs. 45.8%, p<0.001) GI endoscopies (S4 Table).

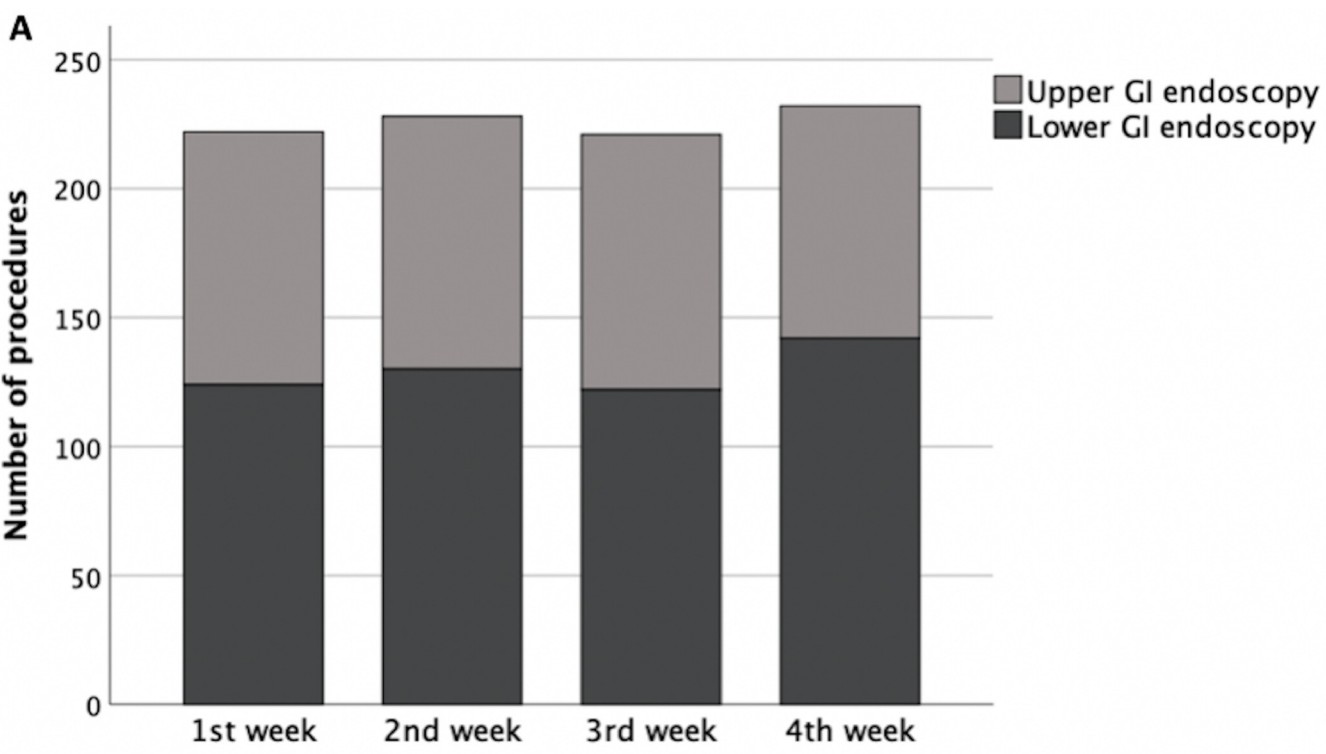

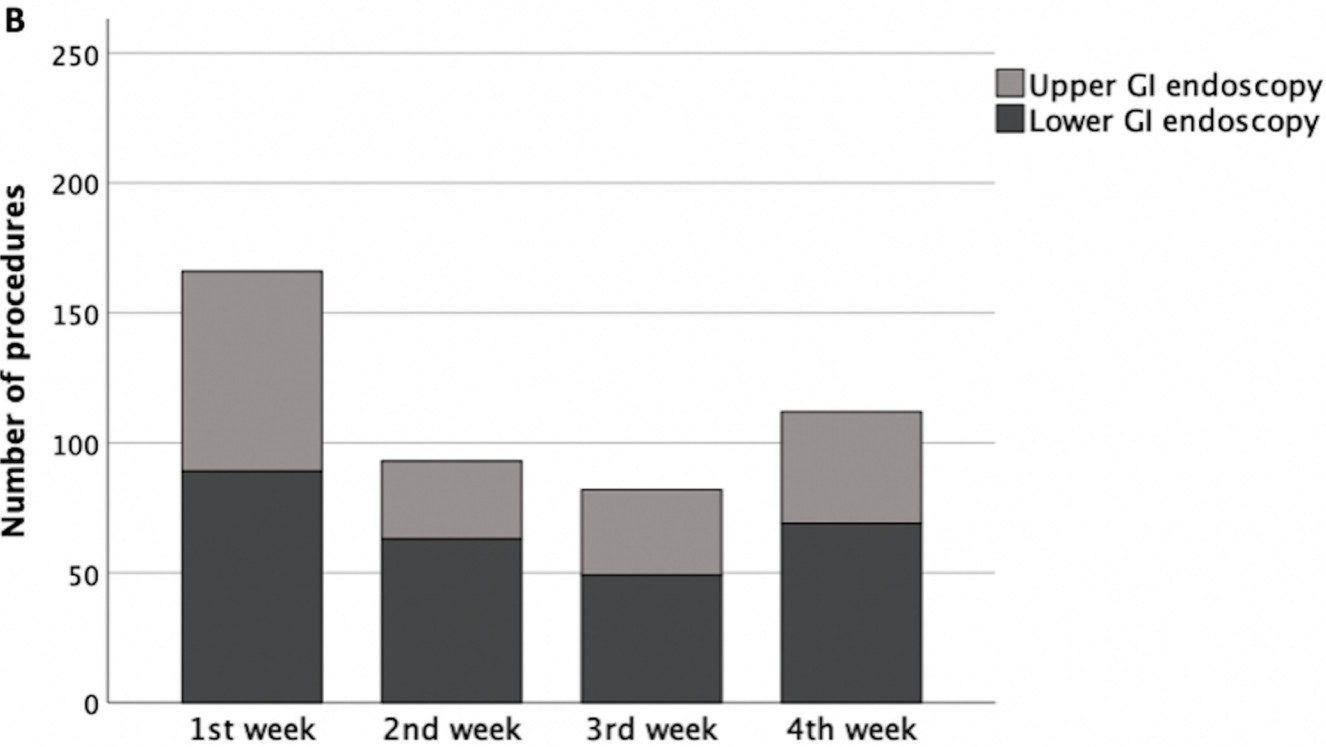

**Fig 1.** Number and type of endoscopies according to gastrointestinal (GI) upper and lower procedures performed in the pre-lockdown (A) and lockdown (B) cohorts.

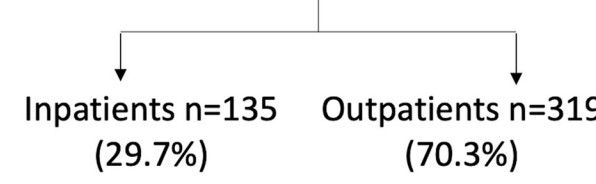

**Pre-alarm cohort**

January 13, 2020 to February 9, 2020

913 endoscopy procedures performed

Colonoscopy, n=471 (51.6%)
Rectosigmoidoscopy, n=42 (4.6%)
Lower EUS, n=4 (0.4%)
Gastroscopy, n=318 (34.8%)
ERCP, n=25 (2.7%)
Upper EUS, n=38 (4.2%)
PEG, n=5 (0.5%)
Others (endoscopic videocapsule, enteroscopy), n=10 (1.1%)

Inpatients n=196 (21.5%)    Outpatients n=717 (78.5%)

**Alarm cohort**

March 16, 2020 to April 14, 2020

454 endoscopy procedures performed

Colonoscopy, n=253 (55.7%)
Rectosigmoidoscopy, n=17 (3.7%)
Lower EUS, n=0 (0%)
Gastroscopy, n=124 (27.3%)
ERCP, n=23 (5.1%)
Upper EUS, n=35 (7.7%)
PEG, n=1 (0.2%)
Others (endoscopic videocapsule, enteroscopy), n=1 (0.2%)

Inpatients n=135 (29.7%)    Outpatients n=319 (70.3%)

**Fig 2. Flow-chart with detailed type of endoscopy procedures performed in the pre-lockdown and lockdown cohorts.** ERCP: Endoscopic retrograde cholangiopancreatography, EUS: Endoscopic ultrasound, PEG: Percutaneous endoscopic gastrostomy.

### Effectiveness of endoscopy activity

The percentage of relevant endoscopic findings were significantly higher in the lockdown cohort compared to the pre-lockdown cohort (31.7% vs. 23.3%, p = 0.006; RR = 1.33, 95% CI: 1.10–1.61) (Table 2, and specific findings in S5 Table). There were also significant differences in the rate of advanced neoplasia/cancer detection rate (18.2% vs. 11.4%, p = 0.004; RR = 1.42, 95% CI: 1.14–1.78) and cancer (8.2% vs. 4%, p = 0.010; RR = 1.58, 95% CI: 1.18–2.13). Table 3 shows all findings according to the upper and lower GI procedures.

A higher percentage of relevant lesions were observed among high-priory indications in the lockdown cohort compared to the pre-lockdown cohort (65.3% vs. 51.5%, p = 0.031; RR = 1.44, 95% CI: 1.03–2.00). There were no significant differences in the percentage of advanced neoplasia/cancer (79.3% vs. 75.6%, p = 0.685; RR = 1.14, 95% CI: 0.69–1.87) and cancer (80.8% vs. 82.7%, p = 1.000; RR = 0.93, 95% CI: 0.47–1.87) in the ESGE high-priority endoscopy procedures. The percentages of advanced neoplasia/cancer and cancer that were observed under ESGE low-priority indications, were 18.6% and 9.1% respectively, without differences between pre-lockdown and lockdown cohorts (Table 4).

### Non-scheduled visits to the hospital

During the lockdown period, 319 (42.5%) outpatient GI endoscopies out of 751 scheduled appointments were performed. 432 scheduled subjects missed the appointment or were

**Table 1. Clinical indications categorized according to the ESGE criteria of outpatient endoscopy procedures in the pre-lockdown and lockdown cohorts.**

| | Pre-lockdown cohort (n = 715*) | Lockdown cohort (n = 319) | p |
|---|---|---|---|
| **High-priority criteria, n (%)** | **327 (45.6%)** | **198 (62.1%)** | **<0.001** |
| Therapeutic endoscopy | 6 (0.8%) | 2 (0.6%) | 1 |
| PEG | 4 (0.6%) | 0 | 0.317 |
| Dysphagia or dyspepsia with alarm symptoms | 64 (9%) | 20 (6.3%) | 0.175 |
| Upper GI bleeding | 4 (0.6%) | 1 (0.3%) | 1 |
| Rectal bleeding | 52 (7.3%) | 21 (6.6%) | 0.793 |
| Colonoscopy for melena after negative upper GI endoscopy | 0 | 1 (0.3%) | 0.309 |
| Severe anemia | 29 (4.1%) | 18 (5.6%) | 0.262 |
| Biopsy for pathology assessment | 10 (1.4%) | 5 (1.6%) | 0.786 |
| Positive FOBT | 119 (16.7%) | 99 (31%) | <0.001 |
| Radiologic evidence of mass | 26 (3.7%) | 22 (6.9%) | 0.026 |
| Pancreatic mass | 13 (1.8%) | 9 (2.8%) | 0.352 |
| **Low-priority criteria, n (%)** | **328 (45.8%)** | **90 (28.2%)** | **<0.001** |
| Endoscopic variceal ligation | 8 (1.1%) | 5 (1.6%) | 0.555 |
| Iron deficiency anemia | 41 (5.8%) | 15 (4.7%) | 0.554 |
| Achalasia | 1 (0.1%) | 0 | 1 |
| Surveillance for Barrett, gastric atrophy and IBD | 60 (8.4%) | 17 (5.3%) | 0.095 |
| Post-endoscopic resection, surgical resection and post-polypectomy surveillance | 124 (17.4%) | 36 (11.3%) | 0.012 |
| Hereditary syndromes | 15 (2.1%) | 1 (0.3%) | 0.030 |
| IBS-like symptoms | 32 (4.5%) | 10 (3.1%) | 0.394 |
| Reflux-disease and dyspepsia without alarm symptoms | 25 (3.5%) | 0 | <0.001 |
| Screening in high-risk patients for cancer | 17 (2.4%) | 6 (1.9%) | 0.820 |
| **Not classifiable, n (%)** | **62 (8.6%)** | **31 (9.7%)** | **0.559** |

PEG: Percutaneous endoscopic gastrostomy, GI: Gastrointestinal, FOBT: Fecal occult blood test, IBD: Inflammatory bowel disease, IBS: Irritable bowel syndrome

*There were 2 outpatient procedures in the pre-lockdown cohort with non-registered indications.

cancelled by our unit, none due to COVID-19 infection. The percentage of patients that required visits to the Emergency Department (3.8% vs. 0.9%, p = 0.01) or hospital admission (3.1% vs. 0.9%, p = 0.031) were higher among attendees (n = 319) compared to no attendees (n = 432) during a mean of 9.1 ± 0.2 months from the scheduled endoscopy date. Importantly, 50% of patients that required visits to the Emergency Department or hospital admission were under high-priority indications; specifically, severe anemia accounted for 23.1% and 25% of cases in each group respectively.

## Safety of performing endoscopic activity during COVID-19 pandemic lockdown

One patient attended for a gastroscopy due to dysphagia (7 days after the procedure), and 2 patients that did not show up to the appointment tested positive for coronavirus by RT-PCR (0.3% vs. 0.5%, p = 0.999). Specifically, there were no more cases detected of symptomatic COVID-19 infection in high GI procedures (1.1% vs. 0%, p = 0.502).

Two physicians (endoscopists) out of 21, tested RT-PCR positive for coronavirus, with symptoms starting on March 10, 2020 and March 19, 2020 (endoscopists number 9 and 15, respectively); neither physician required hospitalization (S1 Fig). The remaining physicians and nurses tested negative for IgG and IgM, except for one asymptomatic nurse (IgG positive). Epidemiological monitoring and testing of close relatives showed that her daughter and

**Table 2. Relevant endoscopic findings according to the ESGE high and low-priority criteria observed in the explorations of the pre-lockdown and lockdown cohorts.**

| | Pre-lockdown cohort (n = 167) | Lockdown cohort (n = 101) | p |
|---|---|---|---|
| **Relevant endoscopic findings among high-priority criteria, n (%)** | **86 (51.5%)** | **66 (65.3%)** | **0.031** |
| Therapeutic endoscopy | 2 (1.2%) | 0 | |
| PEG | 0 | - | |
| Dysphagia or dyspepsia with alarm symptoms | 4 (2.4%) | 5 (5%) | |
| Upper GI bleeding | 1 (0.6%) | 0 | |
| Rectal bleeding | 8 (4.8%) | 8 (7.9%) | |
| Colonoscopy for melena after negative upper GI endoscopy | - | 0 | |
| Severe anemia | 7 (4.2%) | 6 (5.9%) | |
| Biopsy for pathology assessment | 7 (4.2%) | 3 (3%) | |
| Positive FOBT | 41 (24.6%) | 35 (34.7%) | |
| Radiologic evidence of mass | 6 (3.6%) | 5 (5%) | |
| Pancreatic mass | 10 (6%) | 4 (4%) | |
| **Relevant endoscopic findings among low-priority criteria, n (%)** | **67 (40.1%)** | **19 (18.9%)** | **<0.001** |
| Endoscopic variceal ligation | 5 (3%) | 1 (1%) | |
| Iron deficiency anemia | 8 (4.8%) | 3 (3%) | |
| Achalasia | 0 | - | |
| Surveillance for Barrett, gastric atrophy and IBD | 21 (12.6%) | 7 (6.9%) | |
| Post-endoscopic resection, surgical resection and post-polypectomy surveillance | 21 (12.6%) | 8 (7.9%) | |
| Hereditary syndromes | 1 (0.6%) | 0 | |
| IBS-like symptoms | 4 (2.4%) | 0 | |
| Reflux-disease and dyspepsia without alarm symptoms | 6 (3.6%) | - | |
| Screening in high-risk patients for cancer | 1 (0.6%) | 0 | |
| **Relevant endoscopic findings among not classifiable, n (%)** | **14 (8.4%)** | **16 (15.8%)** | **0.073** |

PEG: Percutaneous endoscopic gastrostomy, GI: Gastrointestinal, FOBT: Fecal occult blood test, IBD: Inflammatory bowel disease, IBS: Irritable bowel syndrome

husband were also positive, and infection with compatible symptoms was suspected to take place 7 days before the lockdown.

## Burden of COVID-19

A significant change in the COVID-19 burden was observed in our tertiary hospital during the study period with regards to the number of admissions related to COVID-19, and the total

**Table 3. Endoscopic findings in each cohort according to upper and lower gastrointestinal (GI) procedures.**

| Type of finding | Pre-lockdown cohort | Lockdown cohort | p |
|---|---|---|---|
| **Upper GI endoscopy** | **n = 259** | **n = 92** | |
| Relevant findings | 66 (25.5%) | 34 (37%) | 0.044 |
| Advanced neoplasia and cancer | 15 (5.8%) | 13 (14.1%) | 0.023 |
| Cancer | 13 (5%) | 11 (12%) | 0.031 |
| **Lower GI endoscopy** | **n = 454** | **n = 227** | |
| Relevant findings | 98 (21.6%) | 67 (29.5%) | 0.029 |
| Advanced neoplasia and cancer | 67 (14.8%) | 45 (19.8%) | 0.100 |
| Cancer | 16 (3.5%) | 15 (6.6%) | 0.080 |

GI: Gastrointestinal

**Table 4. Advanced neoplasia/cancer (A) and cancer (B) according to the ESGE high and low-priority indications.**

**A**

|  | Pre-lockdown cohort n = 82 | Lockdown cohort n = 58 | p |
|---|---|---|---|
| **High-priority criteria, n (%)** | **62 (75.6%)** | **46 (79.3%)** | **0.685** |
| Therapeutic endoscopy | 2 (2.4%) | 0 |  |
| PEG | 0 | - |  |
| Dysphagia or dyspepsia with alarm symptoms | 0 | 1 (1.7%) |  |
| Upper GI bleeding | 0 | 0 |  |
| Rectal bleeding | 6 (7.3%) | 5 (8.6%) |  |
| Colonoscopy for melena after negative upper GI endoscopy | - | 0 |  |
| Severe anemia | 1 (1.2%) | 2 (3.4%) |  |
| Biopsy for pathology assessment | 5 (6.1%) | 2 (3.4%) |  |
| Positive FOBT | 35 (42.7%) | 28 (48.3%) |  |
| Radiologic evidence of mass | 4 (4.9%) | 5 (8.6%) |  |
| Pancreatic mass | 9 (11%) | 3 (5.2%) |  |
| **Low-priority criteria, n (%)** | **18 (21.9)** | **8 (13.8%)** | **0.273** |
| Endoscopic variceal ligation | 0 | 0 |  |
| Iron deficiency anemia | 1 (1.2%) | 3 (5.2%) |  |
| Achalasia | 0 | - |  |
| Surveillance for Barrett, gastric atrophy and IBD | 1 (1.2%) | 0 |  |
| Post-endoscopic resection, surgical resection and post-polypectomy surveillance | 14 (17.1%) | 5 (8.6%) |  |
| Hereditary syndromes | 0 | 0 |  |
| IBS-like symptoms | 2 (2.4%) | 0 |  |
| Reflux-disease and dyspepsia without alarm symptoms | 0 | - |  |
| Screening in high-risk patients for cancer | 0 | 0 |  |
| **Not classifiable, n (%)** | **2 (2.4%)** | **4 (6.9%)** | **0.232** |

**B**

|  | Pre-lockdown cohort n = 29 | Lockdown cohort n = 26 | p |
|---|---|---|---|
| **High-priority criteria, n (%)** | **24 (82.7%)** | **21 (80.8%)** | **1.000** |
| Therapeutic endoscopy | 0 | 0 |  |
| PEG | 0 | - |  |
| Dysphagia or dyspepsia with alarm symptoms | 0 | 1 (3.8%) |  |
| Upper GI bleeding | 0 | 0 |  |
| Rectal bleeding | 2 (6.9%) | 1 (3.8%) |  |
| Colonoscopy for melena after negative upper GI endoscopy | - | 0 |  |
| Severe anemia | 0 | 2 (7.7%) |  |
| Biopsy for pathology assessment | 5 (17.2%) | 1 (3.8%) |  |
| Positive FOBT | 5 (17.2%) | 9 (34.6%) |  |
| Radiologic evidence of mass | 3 (10.3%) | 4 (15.4%) |  |
| Pancreatic mass | 9 (31%) | 3 (11.5%) |  |
| **Low-priority criteria, n (%)** | **3 (10.4%)** | **2 (7.7%)** | **1.000** |
| Endoscopic variceal ligation | 0 | 0 |  |
| Iron deficiency anemia | 1 (3.4%) | 2 (7.7%) |  |
| Achalasia | 0 | - |  |
| Surveillance for Barrett, gastric atrophy and IBD | 0 | 0 |  |
| Post-endoscopic resection, surgical resection and post-polypectomy surveillance | 2 (6.9%) | 0 |  |
| Hereditary syndromes | 0 | 0 |  |
| IBS-like symptoms | 0 | 0 |  |
| Reflux-disease and dyspepsia without alarm symptoms | 0 | - |  |

*(Continued)*

**Table 4.** (Continued)

| | | | |
|---|---|---|---|
| Screening in high-risk patients for cancer | 0 | 0 | |
| **Not classifiable, n (%)** | **2 (6.9%)** | **3 (11.5%)** | **0.659** |

PEG: Percutaneous endoscopic gastrostomy, GI: Gastrointestinal, FOBT: Fecal occult blood test, IBD: Inflammatory bowel disease, IBS: Irritable bowel syndrome

number of admissions in the Emergency Department and in the Gastroenterology and Hepatology Department. There was a significant negative correlation between the number of admissions in the Gastroenterology and Hepatology Department and related to COVID-19 (Rho spearman = -0.73, p<0.001; Fig 3). Furthermore, the COVID-19 bed occupation was shown to have peaked (27%) on April 9, 2020.

## Discussion

Our study shows a high detection rate for relevant endoscopic findings in outpatient endoscopy procedures during the COVID-19 pandemic lockdown when ESGE high-risk criteria are considered for prioritization. A low risk of infection was observed in relation to the endoscopy procedure when protection measures were adopted.

The Spanish government declared a "State of Alarm" and lockdown on March 16, 2020 with 9.191 confirmed cases, of which, 75 were on our island and 32 were admitted to our tertiary hospital at that date. At this point, it was essential to reorganize endoscopic activities to protect patients and healthcare workers, and release resources to deal with the COVID-19 burden.

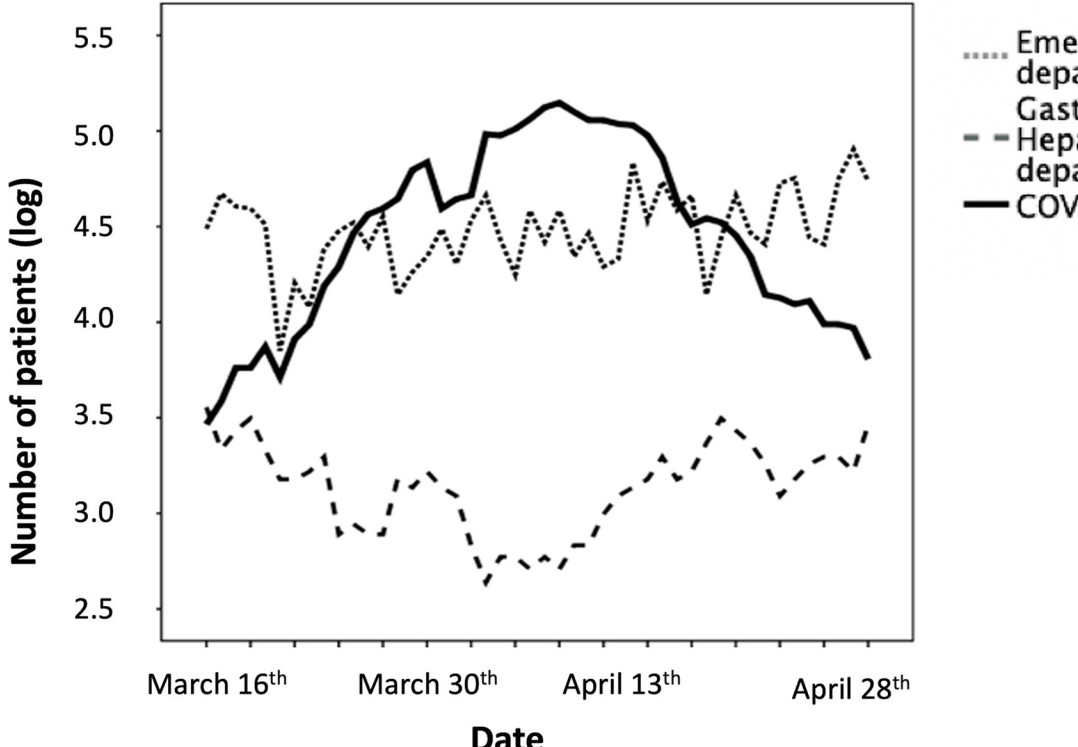

**Fig 3. Burden on healthcare in our tertiary referral hospital during the COVID-19 lockdown.**

Some GI organizations advised triage for the ongoing activity or even the cancellation of non-urgent explorations, subject to region-related situations. In our department, we globally reduced almost 50% of activity homogeneously during the study period according to the local burden of COVID-19 to healthcare resource, which was ten times higher than the activity of the center with the lowest reported activity in our country [21]. According to several published country surveys on endoscopy activity, including more than 500 endoscopy units, the case volume has drastically declined in the majority of centers based on widespread restrictions on elective procedures. Our results of endoscopic activity are in accordance with those of at least 24%-30% of endoscopy units depending on the risk areas of COVID-19 [22–24].

We focused on the evaluation of the appropriateness and safety of outpatient endoscopy procedures due to the urgent indication of the procedures in inpatients, independent of the COVID-19 burden. Indeed, there was a small decrease in the number of inpatient procedures during the lockdown period compared with the pre-lockdown cohort, and most of them showing a high-priority indication (more than 79%). In contrast, the was a significant decrease in inpatient low-priority endoscopies.

According to our results, there was an increase in the number of outpatient high-priority indications, resulting in a 16% increase compared to the pre-lockdown cohort since the onset of the State of Alarm. The increase was particularly notable for positive FOBT and radiologic suspicion of mass indications, which may explain the high number of relevant findings; the highest percentage of relevant findings was under positive FOBT indication. This fact suggests that this indication for colonoscopy should not be omitted. It is also important to note that cancer diagnosis was doubled in the lockdown cohort, although the rate of colorectal cancer could have been higher depending on the positivity threshold of the FOBT in addition to sex and age. Thus fecal hemoglobin concentration might be used to prioritize patients in this rapidly changing scenario [25, 26].

Overall the rate of high-priority indications may be considered suboptimal taking into consideration the circumstances involved in the lockdown. In this regard, if other guideline recommendations were to be considered (i.e., iron deficiency anemia and band-ligation as high priority indications), the percentage could be raised. Another attributable reason is that patients with low-priority indications attended their previously scheduled GI endoscopy procedure which could not be cancelled. In addition, it is important to note the high number of patients that did not show up to the appointments, probably because the fear of being infected overtakes the perceived risk of postponing the procedure. Therefore, counselling and the provision of information on the real risks of contagion versus the delay of procedures should be considered [27].

We selected the ESGE criteria to classify medical indications since they provided detailed high priority indications with the need to be performed either immediately or within 12 weeks in this current scenario. It is important to note that we classified according to ESGE criteria retrospectively due to the guideline release date. Previously, we were using NICE and AEG criteria as regular prioritization in our unit. The ESGE criteria for prioritizing GI procedures proved to be effective, as the majority of significant lesions were diagnosed under high-priority indications. This was especially true for advanced neoplasia/cancer, which substantially increased in the lockdown cohort. A recent published study carried out by our group assessed the usefulness of different prioritizing criteria (NICE, ESGE and Spanish Endoscopic Society criteria) in 1222 outpatient endoscopies during the resumption of endoscopic activity phase [28]. This study also showed a higher risk of significant lesions in the high-priority groups whatever the criteria used, validating the results of the present study. We also evaluated the specific indications among high and low-priority ESGE classification, even non-classifiable but relevant indications, so these results may guide other centers to prioritize endoscopy

procedures in pandemic outbreaks, lockdowns or future waves of COVID-19 pandemic. It is important to highlight that there was a percentage of advanced neoplasia/cancer (18.6%) and cancer (9.1%) cases that were observed under ESGE low-priority indications (post-endoscopic resection, surgical resection and post-polypectomy surveillance, and mild and moderate iron-deficiency anemia). Nevertheless, out of the context of the pandemic, these could be considered as high-priority indications [29, 30]. In our cohort, taking into account these indications as high-priority indications we could have detected 88.5% (n = 23) of these advanced neoplasia/cancer lessions and 100% (n = 5) of cancers.

The rate of non-scheduled visits to the hospital after the endoscopy date was higher among attendees. This probably reflects the endoscopy findings and severity of the disease that justifies hospital assitance. However, half of patients that required visits to the Emergency Department or hospital admission were under high-priority indications supporting the appropriateness of the indication.

Endoscopy units are considered high-risk scenarios for COVID-19 contagion. The potentially risky exposure of the endoscopy staff contrasts with currently available evidence [15]. In our cohort, two physicians with part-time endoscopy activity tested positive after showing symptoms close to the time when protective measures were adopted by our endoscopy unit. In support of the low infection risk, once protective measures were adopted, the screening conducted in our endoscopy unit revealed that only one nurse tested positive for IgG, ruling out asymptomatic cases among the staff.

Endoscopy procedures were equally safe for patients, as the rate of symptomatic infections were similar regarding attendees and non-attendees. However, it should be considered that no positive cases had to undergo endoscopic procedures during the study period. Fortunately, this has been the case for most of the centers (66%) as shown by previously published surveys [22–24]. From the release of guidelines until the peak of infection, our center had to prioritize the available resources and FFP2 masks were unavailable at this point. Nevertheless, all personnel inside the endoscopy room were wearing face shields and surgical masks in addition to the recommended dress code [31], and our endoscopy unit complied with most of the protective measures recommended by most guidelines, which may influence the results [32].

Our study showed results with references to the local burden of COVID-19 infection. These specific data allow for comparisons and extrapolation of our methods and results to other endoscopy units.

This retrospective study has some limitations. First, approximately 50%–70% of subjects are asymptomatic carriers [33], thus, we may have underestimated the number of nosocomial infections as only symptomatic cases were screened with RT-PCR testing and there was not an established screening for patients before or after the procedure. However, this issue was likely to have occurred in both groups. Second, we found a relatively small number of non-scheduled visits to the hospital among included patients, and these data should be taken with caution. Without a doubt, the long term impact evaluation of the delay in GI endoscopy procedures is required with more follow-up time, particularly in colorectal cancer cases [34]. Thirdly, the procedures of non-attenders could not be classified in postponed and missed endoscopy appointments, and unfortunately, clinical indications of non-attenders subjects were not registered due to the lack of registration in the electronic medical record. Finally, the ESGE criteria must be prospectively validated. However, this may be particularly difficult as lockdown scenarios are infrequent which makes our study singular. Nonetheless, our results are based on real-world experience and should be considered until additional evidence is available.

In conclusion, our results argue for the flexibility of endoscopy units to decide the endoscopic activity on a case-by-case basis, preferably guided by high-priority criteria and depending on the prevalence of COVID-19 and available local resources. A balance between

preserving healthcare workers and patients from infection and providing essential services to patients to avoid delays in diagnosis is particularly important during pandemic lockdown.

## Supporting information

**S1 Fig.** COVID-19 infected endoscopists (*) and number of endoscopy procedures by all endoscopists during the pre-lockdown cohort (A) and lockdown cohort (B).
(DOCX)

**S1 Table. Recommendations for gastrointestinal procedures during the COVID-19 pandemic and protective measures adopted by our endoscopy unit.**
(DOCX)

**S2 Table. List of endoscopic findings considered to be relevant.** IBD: Inflammatory bowel disease, GAVE: Gastric antral vascular ectasia.
(DOCX)

**S3 Table. Clinical indications categorized according to the ESGE criteria of inpatient endoscopy procedures in the pre-lockdown and lockdown cohorts.** PEG: Percutaneous endoscopic gastrostomy, GI: Gastrointestinal, FOBT: Fecal occult blood test, IBD: Inflammatory bowel disease, IBS: Irritable bowel syndrome.
(DOCX)

**S4 Table. A.** Clinical indications of outpatient gastrointestinal (GI) upper endoscopy procedures in the pre-lockdown and lockdown cohorts categorized according to ESGE criteria. **B.** Clinical indications of outpatient GI lower endoscopy procedures in the pre-lockdown and lockdown cohorts categorized according to ESGE criteria. PEG: Percutaneous endoscopic gastrostomy, GI: Gastrointestinal, IBD: Inflammatory bowel disease, IBS: Irritable Bowel Syndrome, FOBT: Fecal occult blood test.
(DOCX)

**S5 Table. Relevant endoscopic finding in the pre-lockdown and lockdown cohorts.** IBD: Inflammatory bowel disease, GAVE: Gastric antral vascular ectasia.
(DOCX)

## Acknowledgments

We thank all staff members of the Endoscopy Unit for their vocation, commitment and willingness to cooperate in this health crisis, as well as CIBICAN for editorial support.

## Author Contributions

**Conceptualization:** Dalia Morales-Arraez, David Nicolás-Pérez, Antonio Zebenzui Gimeno-García, Manuel Hernández-Guerra.

**Data curation:** Dalia Morales-Arraez, Anjara Hernández, Alberto Hernández-Bustabad, Carla Amaral, Cristina Reygosa.

**Formal analysis:** Dalia Morales-Arraez.

**Funding acquisition:** Manuel Hernández-Guerra.

**Investigation:** Dalia Morales-Arraez, Manuel Hernández-Guerra.

**Methodology:** Dalia Morales-Arraez, David Nicolás-Pérez, Antonio Zebenzui Gimeno-García, Manuel Hernández-Guerra.

**Project administration:** Dalia Morales-Arraez, Manuel Hernández-Guerra.

**Resources:** Manuel Hernández-Guerra.

**Supervision:** Dalia Morales-Arraez, Manuel Hernández-Guerra.

**Writing – original draft:** Dalia Morales-Arraez.

**Writing – review & editing:** Antonio Zebenzui Gimeno-García, Manuel Hernández-Guerra.

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
