## [Decision Letter · Decision Letter 0]

19 Nov 2021

PONE-D-21-12011

Appropriateness of high-priority criteria and safety of endoscopy procedures during the COVID-19 lockdown

PLOS ONE

Dear Dr. Hernandez-Guerra,

Thank you for submitting your manuscript to PLOS ONE. After careful consideration, we feel that it has merit but does not fully meet PLOS ONE’s publication criteria as it currently stands. Therefore, we invite you to submit a revised version of the manuscript that addresses the points raised during the review process.

We look forward to receiving your revised manuscript.

Kind regards,

Hsu-Heng Yen

Academic Editor

PLOS ONE

Journal Requirements:

Reviewers' comments:

Reviewer's Responses to Questions

**Comments to the Author**

1. Is the manuscript technically sound, and do the data support the conclusions?

Reviewer #1: Yes

Reviewer #2: Yes

2. Has the statistical analysis been performed appropriately and rigorously? 

Reviewer #1: Yes

Reviewer #2: Yes

3. Have the authors made all data underlying the findings in their manuscript fully available?

Reviewer #1: Yes

Reviewer #2: Yes

4. Is the manuscript presented in an intelligible fashion and written in standard English?

Reviewer #1: Yes

Reviewer #2: Yes

5. Review Comments to the Author

Reviewer #1: This is an interesting study. Although most of the countries were under impact of COVID-19 pandemic situation, some of the area had not suffered the serious condition such as east Asia countries. The study can help those area further management before outbreak and lock down.

Reviewer #2: I read with great interest this study, aimed at evaluating the appropriateness of endoscopy procedures during the COVID-19 pandemic lockdown as the rate of high-priority procedures according to the ESGE criteria. Furthermore, the authors evaluated the effectiveness (rate of relevant endoscopic findings, advanced neoplasia/cancer and cancer detection), non-scheduled visits to the hospital and patients and endoscopy staff safety.

By comparing 2 different cohort (lockdown cohort vs pre-lockdown cohort), there was a 49.7% reduction in procedures compared to the pre-lockdown cohort. However, there was a significant increase in the percentage of high-priority indications in the lockdown cohort compared to the pre-lockdown cohort. Similarly, a higher percentage of relevant lesions were observed among high-priory indications in the lockdown cohort compared to the pre-lockdown cohort

The study is well written and well conducted, tables and figures are clear and explicative.

I have only minor comments that I believe could improve the manuscript.

No data have been reported on urgent/emergency endoscopy (i.e. for gastrointestinal bleeding or foreign bodies). In my mind, elective endoscopic activities have been deeply reduced during pandemic, but gastrointestinal urgencies should not be influenced by Covid-19. Do you have data about endoscopy volume in the emergency setting?

In the Introduction section, I would like that the Authors highlight the impact of Covid-19 p on Gastroenterologists and Endoscopists, considering the high-risk procedures they perform (add the reference: Imperatore N, Rispo A, Lombardi G. The price of being a doctor during the COVID-19 outbreak. Gut. 2020;69:1544-1545. doi: 10.1136/gutjnl-2020-321646).

6. PLOS authors have the option to publish the peer review history of their article (what does this mean?). If published, this will include your full peer review and any attached files.

Reviewer #1: No

Reviewer #2: No

---

## [Author Response · Author response to Decision Letter 0]

23 Mar 2022

Reviewer #1: This is an interesting study. Although most of the countries were under impact of COVID-19 pandemic situation, some of the area had not suffered the serious condition such as east Asia countries. The study can help those area further management before outbreak and lock down.

We thank the reviewer for the comment. We agree. Our study showed results with references to the local burden of COVID-19 infection because these specific data allow for comparations and extrapolation of our methods and results to other endoscopy units. We commented this issue in the discussion section (page 23).

Reviewer #2: I read with great interest this study, aimed at evaluating the appropriateness of endoscopy procedures during the COVID-19 pandemic lockdown as the rate of high-priority procedures according to the ESGE criteria. Furthermore, the authors evaluated the effectiveness (rate of relevant endoscopic findings, advanced neoplasia/cancer and cancer detection), non-scheduled visits to the hospital and patients and endoscopy staff safety.

By comparing 2 different cohort (lockdown cohort vs pre-lockdown cohort), there was a 49.7% reduction in procedures compared to the pre-lockdown cohort. However, there was a significant increase in the percentage of high-priority indications in the lockdown cohort compared to the pre-lockdown cohort. Similarly, a higher percentage of relevant lesions were observed among high-priory indications in the lockdown cohort compared to the pre-lockdown cohort

The study is well written and well conducted, tables and figures are clear and explicative.

I have only minor comments that I believe could improve the manuscript.

We thank the reviewer for the time to review and the comments to improve the manuscript. We have updated the main text and the references section with the suggested modifications.

No data have been reported on urgent/emergency endoscopy (i.e. for gastrointestinal bleeding or foreign bodies). In my mind, elective endoscopic activities have been deeply reduced during pandemic, but gastrointestinal urgencies should not be influenced by Covid-19. Do you have data about endoscopy volume in the emergency setting?

We have added the numbers of urgent procedures in both cohorts in the results section (page 10, figure 2). In addition, we have registered all clinical indications for inpatients/urgent endoscopy in supplementary table 3 (page 11).

We agree with the comment of the reviewer and that is why we focused on the evaluation of the appropriateness and safety on outpatient endoscopy procedures due to the usually unavoidable and mandatory urgent indication nature of the procedures in hospitalized patients independently of COVID-19 local burden (discussion section page 19).

In the Introduction section, I would like that the Authors highlight the impact of Covid-19 p on Gastroenterologists and Endoscopists, considering the high-risk procedures they perform (add the reference: Imperatore N, Rispo A, Lombardi G. The price of being a doctor during the COVID-19 outbreak. Gut. 2020;69:1544-1545. doi: 10.1136/gutjnl-2020-321646).

We have added this reference in the introduction section (ref 10). Thank you for your input.

---

## [Editor Report · Decision Letter 1]

4 Apr 2022

Appropriateness of high-priority criteria and safety of endoscopy procedures during the COVID-19 lockdown

PONE-D-21-12011R1

Dear Dr. Hernandez-Guerra,

We’re pleased to inform you that your manuscript has been judged scientifically suitable for publication and will be formally accepted for publication once it meets all outstanding technical requirements.

Kind regards,

Hsu-Heng Yen

Academic Editor

PLOS ONE
---

## [Editor Report · Acceptance letter]

18 Apr 2022

PONE-D-21-12011R1 

Appropriateness of high-priority criteria and safety of endoscopy procedures during the COVID-19 lockdown 

Dear Dr. Hernandez-Guerra:

I'm pleased to inform you that your manuscript has been deemed suitable for publication in PLOS ONE. Congratulations! Your manuscript is now with our production department. 

Kind regards, 

on behalf of

Dr. Hsu-Heng Yen 

Academic Editor

PLOS ONE